# Sjogren’s Syndrome Presenting with Solely Cutaneous Features

**DOI:** 10.3390/diagnostics11071260

**Published:** 2021-07-14

**Authors:** Sneha Centala, Joyce H. Park, Diana Girnita

**Affiliations:** 1TriHealth Good Samaritan Hospital, Cincinnati, OH 45220, USA; 2Palo Alto Medical Foundation, Fremont, CA 94040, USA; joycehopark@gmail.com; 3University of Cincinnati, Cincinnati, OH 45267, USA; girnitdm@ucmail.uc.edu

**Keywords:** Sjogren’s syndrome, cutaneous manifestation, extra glandular

## Abstract

Sjogren’s syndrome is classically characterized by symptoms of keratoconjunctivitis sicca and xerostomia, secondary to lymphocytic infiltration of the salivary and lacrimal glands. Cutaneous findings of this disease are infrequently discussed and thus rarely considered among patients without the typical symptomatology. However, these patients can develop xerosis, alopecia, vitiligo, papular or nodular lesions, or cutaneous vasculitis. A 56-year-old Asian female presented with intermittent cutaneous erythematous lesions of her bilateral pinna and preauricular areas. Despite initial symptom presentation causing concern for tumid lupus versus cutaneous T cell lymphoma versus relapsing polychondritis, extensive serologic and histopathologic workup eventually indicated a likely diagnosis of Sjogren’s syndrome. This case brings to light that Sjogren’s syndrome is truly a multi-systemic disease and can present with primarily extra glandular cutaneous symptoms. When approaching the workup of a new patient, it is absolutely vital to maintain a broad differential and keep in mind that overlap syndromes among multiple autoimmune diseases do exist as well.

## 1. Introduction

Sjogren’s syndrome (SS) is a well-known autoimmune condition involving lymphocytic infiltration of the salivary and lacrimal glands, commonly resulting in keratoconjunctivitis sicca and xerostomia [1,2]. Moreover, a wide range of extra glandular findings can be seen and, rarely, cutaneous findings such as xerosis, alopecia, vitiligo, papular or nodular lesions, or cutaneous vasculitis [3,4].

Unlike SS, Relapsing polychondritis (RP) is a rare systemic autoimmune condition. It can exist alone or in about 30% of cases, alongside other rheumatologic diseases, as an overlap syndrome [5]. Most commonly, it can be associated with a systemic vasculitis. The only way to diagnose RP is clinically and/or histologically, as there are no confirmatory lab tests [6,7]. It is characterized by recurrent inflammation and eventual degeneration of any cartilaginous tissue in the body.

Even more exceptionally rare is tumid lupus erythematosus (TLE). It is considered a variant of chronic cutaneous lupus erythematosus and is difficult to diagnose [8]. Symptoms are limited to the skin and there are typically no other obvious presenting signs.

Here, we present a case of an individual with a solely cutaneous clinical presentation. Despite initial concern for cutaneous T cell lymphoma, tumid lupus erythematosus, or relapsing polychondritis, her extensive diagnostic workup eventually revealed likely diagnosis of Sjogren’s syndrome.

## 2. Case Presentation

The patient is a 56-year-old Asian female, who initially presented to the dermatologist with a small left preauricular erythematous lesion (Figure 1a). She had no associated pain, pruritus, or purulent or sanguineous drainage. She underwent treatment with doxycycline for 2 weeks, but had no improvement. She was reevaluated by the dermatologist who obtained a skin biopsy of that lesion and prescribed hydrocortisone cream. In a week, the lesion started to fade out and then disappeared. Later that same month, the patient developed some left aural edema (Figure 1b), with no associated pain, erythema, or drainage. She used the same hydrocortisone cream that she used for her facial lesion, which resulted in improvement of the swelling, and she did not contact her dermatologist. However, the following month, her right ear pinna (Figure 2) began to develop warmth and swelling without pain. This swelling persisted despite using the hydrocortisone cream and the patient again contacted the dermatologist. During this visit, the dermatologist obtained an additional laboratory workup, performed a second shave biopsy from her right ear, and discussed the results of the first biopsy. The laboratory workup showed a positive antinuclear antibody (ANA), and so a referral to rheumatology was placed for an additional workup.

The first visit with the rheumatology department occurred three months from initial presentation of symptoms. During this evaluation, the patient had persistent swelling of the right pinna, but the facial rash and left ear swelling were completely resolved. Additional clinical findings included intermittent mild xerophthalmia and xerostomia. She uses artificial eye drops as needed, which provide some relief. She has had two dental implants as well. Supplemental review of systems is only positive for intermittent right shoulder pain. Comprehensive rheumatologic review of systems is negative for history of fevers, chills, night sweats, fatigue, weight changes, hair loss, alopecia, scleritis, episcleritis, uveitis, or iritis. The patient denied any nasal crusting, nasal ulcers, sinusitis, hearing loss, oral ulcers, swollen glands or nodes, hoarseness, or voice changes. She also denied dyspnea, cough, chest pain, pericarditis, pleuritis, photosensitivity, skin changes, rashes, psoriasis, or genital ulcers. The patient never complained of abdominal pain, nausea, emesis, hematemesis, bright red blood per rectum, melena, constipation, inflammatory bowel disease, or hematuria. She never experienced Raynaud’s phenomenon, morning stiffness, myalgias, weakness, joint swelling or pain, anemia, stroke, delirium, or psychosis.

The patient has a past medical history of hyperlipidemia and seasonal allergies. Surgical history includes a lumbar laminectomy nine years ago. No significant family history is known. Her only current medications are simvastatin daily, loratadine daily as needed, and topical hydrocortisone cream as needed.

Pertinent physical exam findings revealed right ear cartilage edema with a well healing biopsy site. There was normal appearing left ear cartilage and bilateral preauricular areas without lesions. There was no alopecia, malar rash, oral ulcers, or tonsillar/pharyngeal congestion. Musculoskeletal exam was remarkable for right shoulder limited range of motion only with internal rotation. Remainder of the physical exam was within normal limits. 

Lab findings are detailed in Table 1. Pertinent positive findings are positive ANA (1:2560) with speckled appearance, positive Rheumatoid factor (RF) (70 IU/mL), positive anti-Ro (SSA) and anti-La (SSB) antibodies, elevated erythrocyte sedimentation rate (ESR), and elevated liver enzymes, which had previously been normal. The remainder of the serologic workup was negative, as listed in the table.

Chest radiograph displays no sign of acute cardiopulmonary disease but does reveal a seven-millimeter nodular density in the right lower lung, that was not confirmed on a subsequent study. Pulmonary function test shows normal spirometry and flow volume loop findings. Echocardiogram demonstrates normal biventricular size and systolic function, normal left ventricular ejection fraction, no significant valvular abnormalities, and normal estimated pulmonary artery systolic pressure.

A punch biopsy was performed of the left preauricular lesion (Figure 3) at the time of presentation to the dermatologist. Histopathology of the tissue biopsy demonstrates atypical lymphoid infiltrate in the superficial and deep dermis with perivascular and adnexal involvement within the follicular epidermis. The overlying epidermis is largely uninvolved. No significant dermal mucin deposition was noted. Immunohistochemical stains highlight a CD3 positive infiltrate with a significant skew of more CD4 than CD8 presence within the follicular epithelium. Additional immunohistochemical staining is negative for CD20, BCL-6, CD10, CD21, PD1, and therefore negative for chronic T cell lymphoma, B cell lymphoma, and T cell receptor clonal rearrangements. The findings indicate a connective tissue disorder, possible tumid lupus.

A shave biopsy was later performed of the right ear helix (Figure 4), about two months after initial presentation. This specimen demonstrates telangiectasia with a superficial perivascular lymphohistiocytic infiltrate. Immunohistochemical panel shows a normal ratio of CD4 and CD8 T cells, without evidence of chronic T cell lymphoma. There is no significant dermal mucin nor increased plasmacytoid dendritic cells, again pointing towards a connective tissue disease pathology. 

The patient’s presenting right ear swelling has improved with use of 2.5% hydrocortisone, within about 2 months from initial presentation. However, about six weeks after her initial visit with rheumatology, she has developed joint pain of her bilateral shoulders, wrists, and knees. She has associated morning stiffness, which lasts about 45 min to 1 h. No joint swelling is present. These symptoms are improved with movement and use of the joints as well as ibuprofen twice daily. She has most recently been initiated on hydroxychloroquine treatment, currently at a dose of 200 mg daily, and she continues to remain free of symptoms at this time. The patient was referred and evaluated further by ophthalmology, which revealed mild dry eyes with signs of corneal abrasions, consistent with findings of SS.

## 3. Discussion

This patient has an extremely unusual clinical presentation in relation to her objective immunologic findings. Her initial presentation with a left preauricular lesion was suspicious for tumid lupus versus a type of T cell lymphoma. However, the sequence of events that included the swelling of the cartilaginous area of the left ear, followed by the swelling of the right ear also raised the suspicion of relapsing polychondritis.

Tumid lupus patients may present with pink or violaceous papules or plaques without other manifestations of systemic lupus erythematosus (SLE). Histopathologically, TLE differs from SLE due to lack of epidermal or dermo-epidermal involvement and classic findings of superficial and deep lymphohistiocytic inflammatory infiltrates in a perivascular and adnexal distribution [9]. There is a prominence of CD3 and CD4 lymphocytes, with a lower percentage of CD8. This distribution distinguishes this disease from others, such as lymphomas, pseudolymphomas, and polymorphous light eruption [10]. There is typically also significant mucin deposition, and the lack thereof in the biopsy sample actually questions if this patient had tumid lupus. 

Cutaneous T cell lymphoma, one of the initially deliberated diagnoses, is actually more of an umbrella diagnosis. It includes various more specific conditions such as mycosis fungoides, Sezary syndrome, adult T cell lymphoma/leukemia, primary cutaneous CD30+ lymphoproliferative disorders, primary cutaneous peripheral T cell lymphoma, and subcutaneous panniculitis like T cell lymphoma. Diagnosis requires histopathologic evidence, supportive immunocytochemistry, and accurate T cell clonality assays [11]. Multiple immunohistochemical stains were performed on the tissue samples for this patient, without sign of a cutaneous T cell lymphoma, further supporting a connective tissue etiology.

When considering RP, she only met two of the criteria defined by McAdam et al. [3], but the response to steroid may have allowed for the diagnosis per criteria of Damiani and Levine [2]. However, this patient had no signs of other cartilaginous involvement including the sinuses, trachea, nasal cartilage, lung, or aorta. The additional serologic workup, which revealed positive ANA, SSA, SSB, and RF confounded that initial potential diagnosis. RP can present with positive ANA, but presence of these other more specific antibodies more likely indicated Sjogren’s syndrome. 

This patient only had mild symptoms of xerostomia and xerophthalmia, and they were never significant enough for her to seek medical workup. She rather presented with the preauricular rashes and cartilaginous edema. These symptoms along with the confounding lab and pathology findings caused some confusion surrounding her final diagnosis. Upon literature review, there have been very few documented presentations of solely the cutaneous manifestations of Sjogren’s syndrome, and none in this specific manner of mimicking RP. The cutaneous findings associated with SS seem to be underestimated in general [8]. It is very important to maintain SS within the rheumatologic differential when approaching patients with new skin lesions, even without the classic keratoconjunctivitis sicca or xerostomia.

## Figures and Tables

**Figure 1 diagnostics-11-01260-f001:**
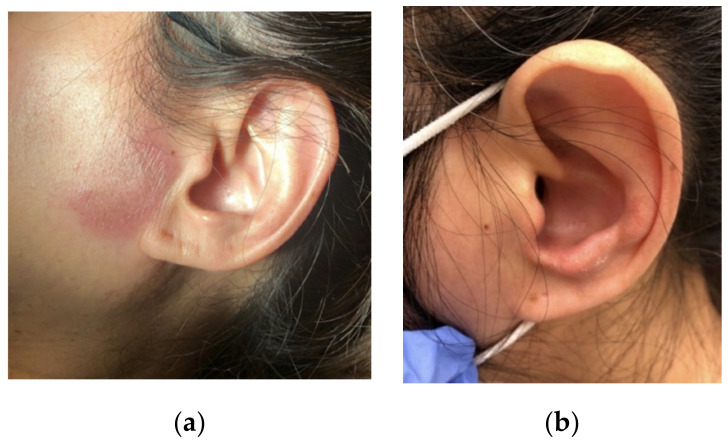
(**a**) Left preauricular lesion. (**b**) Left ear edema.

**Figure 2 diagnostics-11-01260-f002:**
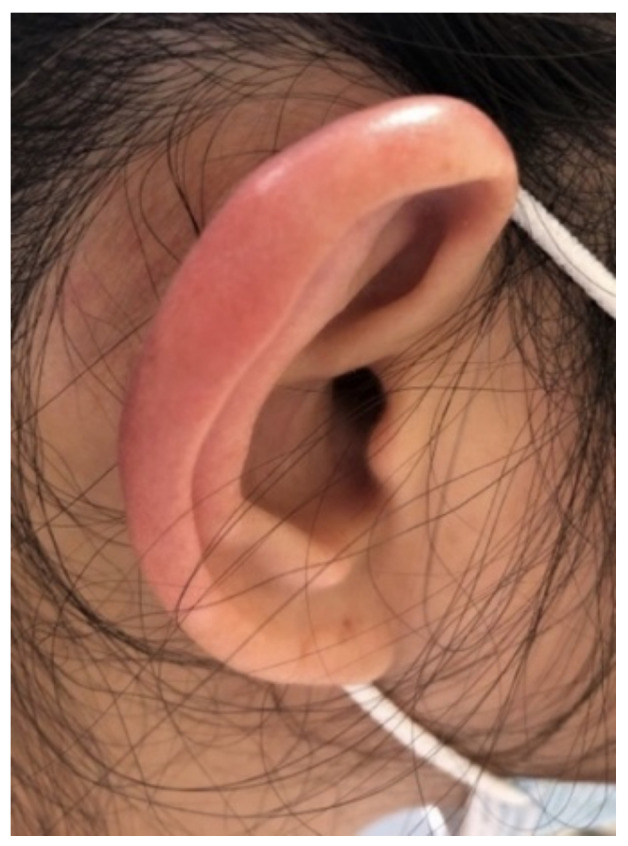
Right ear pinna.

**Figure 3 diagnostics-11-01260-f003:**
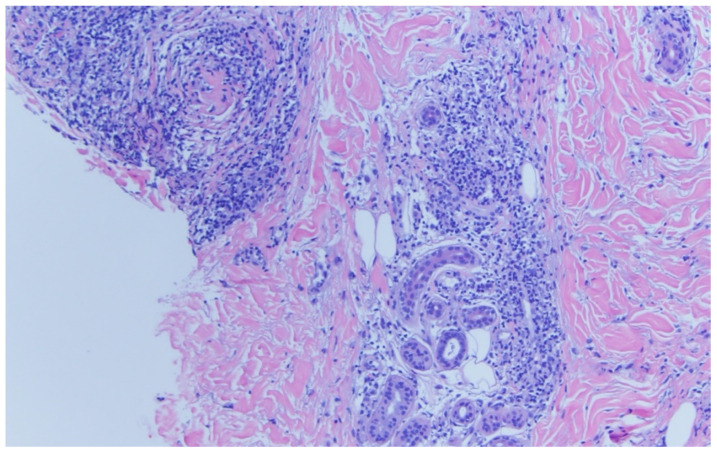
Punch biopsy specimen from left preauricular lesion.

**Figure 4 diagnostics-11-01260-f004:**
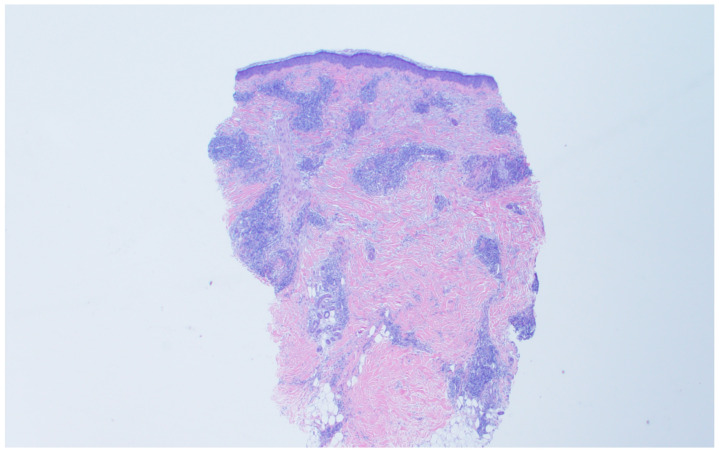
Shave biopsy specimen from right ear helix.

**Table 1 diagnostics-11-01260-t001:** Laboratory findings.

	Lab Finding	Normal Range
**Complete Blood Count**		
White blood cells (WBC)	3.8 × 10^9^ cells/L	4.5–11 × 10^9^ cells/L
Hemoglobin (Hgb)	13.7 g/dL	13.5–17.5 g/dL
Hematocrit (Hct)	42%	41–50 %
**Liver Enzymes**		
Aspartate transaminase (AST)	45 u/L	5–40 u/L
Alanine transaminase (ALT)	63 u/L	19–25 u/L
**Inflammatory Markers**		
Erythrocyte sedimentation rate (ESR)	33 mm/hr	0–30 mm/hr
C-reactive protein (CRP)	2.9 mg/L	<10 mg/L
**Autoimmune Serology**		
Antinuclear antibody (ANA)	1:2560, speckled	Negative, <1:80
Rheumatoid factor (RF)	70 IU/mL	0–20 IU/mL
Anti-Ro antibody (SSA)	Positive	Negative, <20
Anti-La antibody (SSB)	Positive	Negative, <20
Complement 3 (C3)	Within normal range	88–201 mg/dL
Complement 4 (C4)	Within normal range	15–45 mg/dL
Ribosomal P antibody	Negative	Negative, <15 U/mL
Double stranded DNA antibody (dsDNA)	Negative	Negative, <12 U/mL
Sm antibody	Negative	Negative, <7 U/mL
Topoisomerase 1 antibody (Scl 70)	Negative	Negative, <32 U/mL
Thyroid peroxidase antibody (TPO)	Negative	Negative, <16 U/mL
Antineutrophil cytoplasmic antibody (ANCA)	Negative	Negative, <20 IU/mL
Angiotensin converting enzyme (ACE)	Negative	Negative, <40 nmol/mL/min
Antiphospholipid panel	Negative	None detected APLNegative, <12 IgM/IgA
**Urine Analysis**		
Protein	None	None
Red blood cells	None	None
**Serum Protein Electrophoresis**		
Total Protein	7.5 g/dL	6.4–8.3 g/dL
Albumin	4.19 g/dL	3.5–5.0 g/dL
Alpha 1 globulin	0.27 g/dL	0.1–0.3 g/dL
Alpha 2 globulin	0.71 g/dL	0.6–1.0 g/dL
Beta globulin	0.68 g/dL	0.7–1.2 g/dL
Gamma globulin	1.65 g/dL	0.7−1.6 g/dL
Paraprotein	None	None

## Data Availability

Not applicable.

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
