# Peer review of "Sjogren’s Syndrome Presenting with Solely Cutaneous Features"

_diagnostics, 2021, doi:10.3390/diagnostics11071260_

Round 1

Reviewer 1 Report

When there are symptoms such as redness or swelling of the ear auricle, it is necessary to differentially diagnose various infections and dermatitis, etc. in addition to relapsing polychondritis. In this patient, i) other characteristic symptoms of relapsing polychondritis, such as eye involvement, nose involvement, large airway involvement, and joint involvement, were absent, ii) erythematous lesions were also observed on the cheek as well as auricle in figure 1a, Moreover, auricular lesions were not typical for relapsing polychondritis, iii) there were no specific findings suggestive of relapsing polychondritis in the auricular biopsy. Considering these findings, it would be difficult to say that this patient was mimicking relapsing polychondritis just with the atypical auricular skin lesion shown in the report.

Author Response

1) Abstract. This case brings to light that patients with Sjogren’s syndrome can in fact present with extra glandular cutaneous symptoms. When approaching the workup of a new patient, it is always vital to maintain a broad differential and remember that over- lap between multiple autoimmune diseases is also likely. It is important to discuss the multi systemic involvement of this disease. Please ameliorate this paragraph.

Edit: This case brings to light that Sjogren’s syndrome is truly a multi systemic disease and can present with primarily extra glandular cutaneous symptoms. When approaching the workup of a new patient, it is absolutely vital to maintain a broad differential and keep in mind that overlap syndromes among multiple autoimmune diseases do exist as well.

2) Introduction. Here, we present a case of an individual with an interesting clinical presentation and diagnostic workup, to eventually reveal a likely diagnosis of Sjogren’s syndrome with possible concurrent tumid lupus erythematosus, mimicking relapsing polychondritis. Please ameliorate this sentence.

Edit: Here, we present a case of an individual with a solely cutaneous clinical presentation, strongly mimicking relapsing polychondritis. Despite initial concern for cutaneous T cell lymphoma or tumid lupus erythematosus, her extensive diagnostic workup eventually revealed likely diagnosis of Sjogren’s syndrome.

3) Case report. Pertinent physical exam findings include right ear cartilage edema with a well healing biopsy site, normal left ear cartilage, normal preauricular areas without lesions. There is no alopecia, malar rash, oral ulcers, or tonsillar or pharyngeal congestion. Pupils are round and equally reactive to light. Musculoskeletal exam was only remarkable for right shoulder limited range of motion with internal rotation. The rest of the physical exam is within normal limits. Please improve these sentences.

Edit: Pertinent physical exam findings revealed right ear cartilage edema with a well healing biopsy site. There was normal appearing left ear cartilage and bilateral preauricular areas without lesions. There was no alopecia, malar rash, oral ulcers, or tonsillar/pharyngeal congestion. Musculoskeletal exam was remarkable for right shoulder limited range of motion only with internal rotation. Remainder of the physical exam was within normal limits.

4) Case report. Histopathology of the tissue biopsy demonstrates atypical lymphoid infiltrate in the superficial and deep dermis with perivascular and adnexal involvement within the follicular epidermis. Please add some references.

These were findings documented by the pathologist, not found from an outside resource. I am unclear as to any references to include for this?

5) Discussion. This patient did not present with complaints of the classic symptoms of dry eyes, mouth, or skin, although she had mild symptoms that never bothered her enough to  search for a diagnosis. Having primarily the skin involvement with rash in the preauricular area, followed by swelling of the cartilaginous area of the left ear and subsequently  the right ear caused more confusion regarding her diagnosis. Upon literature review, there have been very few documented presentations of solely the cutaneous manifestations of Sjogren’s syndrome, and none in this specific manner, mimicking RP. The cutaneous findings that can be associated with SS seem to be underestimated [8].  It is very important to maintain SS within the rheumatologic differential when approaching patients with new skin lesions, even without the classic keratoconjunctivitis sicca or xerostomia. Could you please improve the conclusion?

Edit: This patient only had mild symptoms of xerostomia and xerophthalmia, and they were never significant enough for her to seek medical workup. She rather presented with the preauricular rashes and cartilaginous edema. These symptoms along with the confounding lab and pathology findings caused some confusion surrounding her final diagnosis. Upon literature review, there have been very few documented presentations of solely the cutaneous manifestations of Sjogren’s syndrome, and none in this specific manner of mimicking RP. The cutaneous findings associated with SS seem to be underestimated in general [8]. It is very important to maintain SS within the rheumatologic differential when approaching patients with new skin lesions, even without the classic keratoconjunctivitis sicca or xerostomia.

6) Figure 2. Shave Biopsy Specimen from Right Ear Helix. Please add a legend.

Description of pathology images added.

7) Comments & Suggestions: When there are symptoms such as redness or swelling of the ear auricle, it is necessary to differentially diagnose various infections and dermatitis, etc. in addition to relapsing polychondritis. In this patient, i) other characteristic symptoms of relapsing polychondritis, such as eye involvement, nose involvement, large airway involvement, and joint involvement, were absent, ii) erythematous lesions were also observed on the cheek as well as auricle in figure 1a, Moreover, auricular lesions were not typical for relapsing polychondritis, iii) there were no specific findings suggestive of relapsing polychondritis in the auricular biopsy. Considering these findings, it would be difficult to say that this patient was mimicking relapsing polychondritis just with the atypical auricular skin lesion shown in the report.

Response: The initial history and physical exam findings were not suspicious for infectious etiology. There had been consideration for dermatitis, but no clear inciting agent or other obvious etiology. I understand this patient did not have any other typical presenting findings of relapsing polychondritis. Would it be more accurate to note relapsing polychondritis as one of the initial considered differentials, along with the cutaneous T cell lymphoma and tumid lupus erythematosus? Rather than say the presentation was fully mimicking RP? It can be more "Sjogren's Syndrome Presenting with Solely Cutaneous Features". 

Reviewer 2 Report

1) Abstract. This case brings to light that patients with Sjogren’s syndrome can in fact present with extra glandular cutaneous symptoms. When approaching the workup of a new patient, it is always vital to maintain a broad differential and remember that over- lap between multiple autoimmune diseases is also likely. It is important to discuss the multi systemic involvement of this disease. Please ameliorate this paragraph.

2) Introduction. Here, we present a case of an individual with an interesting clinical presentation and diagnostic workup, to eventually reveal a likely diagnosis of Sjogren’s syndrome with possible concurrent tumid lupus erythematosus, mimicking relapsing polychondritis. Please ameliorate this sentence.

3) Case report. Pertinent physical exam findings include right ear cartilage edema with a well healing biopsy site, normal left ear cartilage, normal preauricular areas without lesions. There is no alopecia, malar rash, oral ulcers, or tonsillar or pharyngeal congestion. Pupils are round and equally reactive to light. Musculoskeletal exam was only remarkable for right shoulder limited range of motion with internal rotation. The rest of the physical exam is within normal limits. Please improve these sentences.

4) Case report. Histopathology of the tissue biopsy demonstrates atyp ical lymphoid infiltrate in the superficial and deep dermis with perivascular and adnexal involvement within the follicular epidermis. Please add some references.

5) Discussion. This patient did not present with complaints of the classic symptoms of dry eyes, mouth, or skin, although she had mild symptoms that never bothered her enough to  search for a diagnosis. Having primarily the skin involvement with rash in the preauricular area, followed by swelling of the cartilaginous area of the left ear and subsequently  the right ear caused more confusion regarding her diagnosis. Upon literature review, there have been very few documented presentations of solely the cutaneous manifestations of Sjogren’s syndrome, and none in this specific manner, mimicking RP. The cutaneous findings that can be associated with SS seem to be underestimated [8].  It is very important to maintain SS within the rheumatologic differential when approaching patients with new skin lesions, even without the classic keratoconjunctivitis sicca or xerostomia. Could you please improve the conclusion?

6) Figure 2. Shave Biopsy Specimen from Right Ear Helix. Please add a legend.

Author Response

1) Abstract. This case brings to light that patients with Sjogren’s syndrome can in fact present with extra glandular cutaneous symptoms. When approaching the workup of a new patient, it is always vital to maintain a broad differential and remember that over- lap between multiple autoimmune diseases is also likely. It is important to discuss the multi systemic involvement of this disease. Please ameliorate this paragraph.

Edit: This case brings to light that Sjogren’s syndrome is truly a multi systemic disease and can present with primarily extra glandular cutaneous symptoms. When approaching the workup of a new patient, it is absolutely vital to maintain a broad differential and keep in mind that overlap syndromes among multiple autoimmune diseases do exist as well.

2) Introduction. Here, we present a case of an individual with an interesting clinical presentation and diagnostic workup, to eventually reveal a likely diagnosis of Sjogren’s syndrome with possible concurrent tumid lupus erythematosus, mimicking relapsing polychondritis. Please ameliorate this sentence.

Edit: Here, we present a case of an individual with a solely cutaneous clinical presentation, strongly mimicking relapsing polychondritis. Despite initial concern for cutaneous T cell lymphoma or tumid lupus erythematosus, her extensive diagnostic workup eventually revealed likely diagnosis of Sjogren’s syndrome.

3) Case report. Pertinent physical exam findings include right ear cartilage edema with a well healing biopsy site, normal left ear cartilage, normal preauricular areas without lesions. There is no alopecia, malar rash, oral ulcers, or tonsillar or pharyngeal congestion. Pupils are round and equally reactive to light. Musculoskeletal exam was only remarkable for right shoulder limited range of motion with internal rotation. The rest of the physical exam is within normal limits. Please improve these sentences.

Edit: Pertinent physical exam findings revealed right ear cartilage edema with a well healing biopsy site. There was normal appearing left ear cartilage and bilateral preauricular areas without lesions. There was no alopecia, malar rash, oral ulcers, or tonsillar/pharyngeal congestion. Musculoskeletal exam was remarkable for right shoulder limited range of motion only with internal rotation. Remainder of the physical exam was within normal limits.

4) Case report. Histopathology of the tissue biopsy demonstrates atypical lymphoid infiltrate in the superficial and deep dermis with perivascular and adnexal involvement within the follicular epidermis. Please add some references.

Question: I am not sure what references to add for this? These were findings documented by the pathologist, not found from an outside resource.

5) Discussion. This patient did not present with complaints of the classic symptoms of dry eyes, mouth, or skin, although she had mild symptoms that never bothered her enough to  search for a diagnosis. Having primarily the skin involvement with rash in the preauricular area, followed by swelling of the cartilaginous area of the left ear and subsequently  the right ear caused more confusion regarding her diagnosis. Upon literature review, there have been very few documented presentations of solely the cutaneous manifestations of Sjogren’s syndrome, and none in this specific manner, mimicking RP. The cutaneous findings that can be associated with SS seem to be underestimated [8].  It is very important to maintain SS within the rheumatologic differential when approaching patients with new skin lesions, even without the classic keratoconjunctivitis sicca or xerostomia. Could you please improve the conclusion?

Edit: This patient only had mild symptoms of xerostomia and xerophthalmia, and they were never significant enough for her to seek medical workup. She rather presented with the preauricular rashes and cartilaginous edema. These symptoms along with the confounding lab and pathology findings caused some confusion surrounding her final diagnosis. Upon literature review, there have been very few documented presentations of solely the cutaneous manifestations of Sjogren’s syndrome, and none in this specific manner of mimicking RP. The cutaneous findings associated with SS seem to be underestimated in general [8]. It is very important to maintain SS within the rheumatologic differential when approaching patients with new skin lesions, even without the classic keratoconjunctivitis sicca or xerostomia.

6) Figure 2. Shave Biopsy Specimen from Right Ear Helix. Please add a legend.

Added pathology info with each image.

Reviewer 3 Report

The manuscript (ID: diagnostics-1272004, type of manuscript: Case Report) entitled “Sjogren’s Syndrome with Cutaneous Features Mimicking Relapsing Polychondritis?” can be published in Diagnostics - Section: Pathology and Molecular Diagnostics after minor revision. I am not an expert in this rare kind of disease, however, it is not quite clear to me why not taking all information available online and discussing it in the context/ framework of such a case report.

Rovenský J, Sedlácková M. Relabujúca polychondritída [Relapsing polychondritis]. Cas Lek Cesk. 2012;151(2):64-8. Slovak. PMID: 22515011.

Abstract: Relapsing polychondritis (RP) is an unusually rare disease involving multiple organs. It has an episodic course, occasionally also progressing. Typically, inflammation of cartilaginous tissues and tissues rich in glycosaminoglycans is present. Clinical symptoms are concentrated in auricula, nose, larynx, upper respiratory tract, joints, heart, blood vessels, inner ear, cornea and sclera. Manifestations include: (1) chondritis of auricular, nasal, laryngotracheal, costal and joint cartilages, (2) inflammation of the eyes and inner ear, (3) collapse of laryngotracheal structures and structures in the subglottic area resulting in increased susceptibility to upper respiratory tract infections, (4) diversity of clinical manifestations, of the disease course and also of the treatment response. Concurrent systemic vasculitis or glomerulonephritis may contribute to higher morbidity and premature mortality. In about 30% of cases the RP is secondary, accompanied by other systemic connective tissue disorders as RA, SLE, Sjögren's syndrome, thyroiditis, ulcerative colitis, psoriasis and Behćet's syndrome. Diagnosis is based on 1986 diagnostic criteria from Minnesota and RP has to be suspected when the inflammatory bouts involve at least two of the typical sites - auricular, nasal, laryngotracheal or one of the typical sites and two other--ocular, statoacoustic disturbances (hearing loss and/or vertigo) and arthritis. In the treatment are, apart from corticoids and nonsteroidal anti-inflammatory drugs, also corticoids combined with immunosuppressive therapy (cyclophosphamide, azathioprine, chlorambucil, cyclosporine) used. More recently, also biologic therapy is used in RP (infliximab, adalimumab, ethanercept, tocilizumab, rituximab). It is necessary to underscore that biologic therapy for RP is only a research modality used in very severe refractory forms of RP. Preliminary results suggest that biologic therapy will have its place in severe refractory relapsing forms of RP.

Author Response

In the Introduction section, I briefly mentioned some background info about these conditions. Relapsing polychondritis is typically known well enough by the Rheumatology community, so I believe we assumed the great majority of readers would be familiar with the disease and its pathophysiology.

Round 2

Reviewer 1 Report

As the authors agreed, if this patient did not have any other typical presenting findings of relapsing polychondritis, the title of this case report and the sentences such as “strongly mimicking relapsing polychondritis (line 83)“ should be changed. Moreover, the introduction which begins with a description of the RP (lines 68-73), also needs to be modified.

Author Response

Modifications are detailed below:

Title: Sjogren’s Syndrome Presenting with Solely Cutaneous Features

Introduction: Sjogren’s syndrome (SS) is a well-known autoimmune condition involving lymphocytic infiltration of the salivary and lacrimal glands, commonly resulting in keratoconjunctivitis sicca and xerostomia [5,6]. Moreover, a wide range of extra glandular findings can be seen and rarely cutaneous findings such as xerosis, alopecia, vitiligo, papular or nodular lesions, or cutaneous vasculitis [7,8].

Unlike SS, Relapsing polychondritis (RP) is a rare systemic autoimmune condition. It can exist alone or in about 30% of cases, alongside other rheumatologic diseases, as an overlap syndrome [1]. Most commonly, it can be associated with a systemic vasculitis. The only way to diagnose RP is clinically and/or histologically, as there are no confirmatory lab tests [2,3]. It is characterized by recurrent inflammation and eventual degeneration of any cartilaginous tissue in the body.

Even more exceptionally rare is tumid lupus erythematosus (TLE). It is considered a variant of chronic cutaneous lupus erythematosus and is difficult to diagnose [4]. Symptoms are limited to the skin and there are typically no other obvious presenting signs.

Here, we present a case of an individual with a solely cutaneous clinical presentation. Despite initial concern for cutaneous T cell lymphoma, tumid lupus erythematosus, or relapsing polychondritis, her extensive diagnostic workup eventually revealed likely diagnosis of Sjogren’s syndrome.

Case Presentation/Discussion: Minor wording changes were made to reflect that RP was not a strong diagnosis within the differential. 

Reviewer 2 Report

The manuscript has been improved, as required. I have no further comments.

Author Response

Modifications are detailed below, based on the recommendations of all reviewers:

Title: Sjogren’s Syndrome Presenting with Solely Cutaneous Features

Introduction: Sjogren’s syndrome (SS) is a well-known autoimmune condition involving lymphocytic infiltration of the salivary and lacrimal glands, commonly resulting in keratoconjunctivitis sicca and xerostomia [5,6]. Moreover, a wide range of extra glandular findings can be seen and rarely cutaneous findings such as xerosis, alopecia, vitiligo, papular or nodular lesions, or cutaneous vasculitis [7,8].

Unlike SS, Relapsing polychondritis (RP) is a rare systemic autoimmune condition. It can exist alone or in about 30% of cases, alongside other rheumatologic diseases, as an overlap syndrome [1]. Most commonly, it can be associated with a systemic vasculitis. The only way to diagnose RP is clinically and/or histologically, as there are no confirmatory lab tests [2,3]. It is characterized by recurrent inflammation and eventual degeneration of any cartilaginous tissue in the body.

Even more exceptionally rare is tumid lupus erythematosus (TLE). It is considered a variant of chronic cutaneous lupus erythematosus and is difficult to diagnose [4]. Symptoms are limited to the skin and there are typically no other obvious presenting signs.

Here, we present a case of an individual with a solely cutaneous clinical presentation. Despite initial concern for cutaneous T cell lymphoma, tumid lupus erythematosus, or relapsing polychondritis, her extensive diagnostic workup eventually revealed likely diagnosis of Sjogren’s syndrome.

Case Presentation/Discussion: Minor wording changes were made to reflect that RP was not a strong diagnosis within the differential. 

Round 3

Reviewer 1 Report

1. The numbering of references should be revised.